# Aneurysmal subarachnoid hemorrhage in pregnancy: National trends of treatment, predictors, and outcomes

**Kasra Khatibi**[1][⊙], **Hamidreza Saber**[2][⊙], **Smit Patel**[3], **Lucido Luciano Ponce Mejia**[4], **Naoki Kaneko**[5], **Viktor Szeder**[5], **May Nour**[3,5], **Reza Jahan**[5], **Satoshi Tateshima**[5], **Geoffrey Colby**[6], **Gary Duckwiler**[5], **Yalda Afshar**[7]*

1 Department of Neurosurgery, University of Southern California, Los Angeles, CA, United States of America,
2 Department of Neurology, University of Texas at Austin, Austin, TX, United States of America,
3 Department of Neurology, University of California Los Angeles, Los Angeles, CA, United States of America,
4 Department of Neurosurgery, Louisiana State University, New Orleans, LA, United States of America,
5 Department of Radiology, University of California Los Angeles, Los Angeles, CA, United States of America,
6 Department of Neurosurgery, University of California Los Angeles, Los Angeles, CA, United States of America, 7 Department of Obstetrics and Gynecology, University of California Los Angeles, Los Angeles, CA, United States of America

⊙ These authors contributed equally to this work.
* yafshar@mednet.ucla.edu

**Data Availability Statement:** The data underlying the results presented in the study are available from National Inpatient Sample (NIS) database. All data is available for download directly from NIS:

## Abstract

### Introduction

Aneurysmal subarachnoid hemorrhage (aSAH) is a rare event associated with significant pregnancy-associated maternal and neonatal morbidity and mortality. The optimal treatment strategy and clinical outcome of aSAH in pregnancy remains unclear. We aimed to investigate the treatment utilizations and outcomes of aSAH in pregnant people.

### Methods

Using the 2010–2018 National Inpatient Sample, we identified all birth hospitalizations of women between ages of 18 to 45 associated with subarachnoid hemorrhage and aneurysm treatment were included. Multivariate analyses were used to evaluate the effect of pregnancy state, mode of treatment of aneurysms, severity of subarachnoid hemorrhage on mortality and discharge destination of this cohort. Trends in mode of treatment utilized for aneurysmal treatment in this time interval was evaluated.

### Results

13,351 aSAH with treatment were identified, of which 440 were associated with pregnancy. There was no significant difference in mortality or rate of discharge to home in pregnancy related hospitalization. Worse aSAH severity, chronic hypertension, and smaller hospital size was associated with significantly higher rate of mortality from aSAH during pregnancy. Worse aSAH severity was associated with lower rate of discharge to home. Like the non-pregnant cohort, the treatment of ruptured aneurysms in pregnancy are increasingly through

https://hcup-us.ahrq.gov/db/nation/nis/nisdbdocumentation.jsp

**Funding:** The author(s) received no specific funding for this work.

**Competing interests:** The authors have declared that no competing interests exist.

endovascular approaches. The mode of treatment does not change the mortality or discharge destination.

## Conclusions

Pregnancy does not alter mortality or the discharge destination for aSAH. Ruptured aneurysms during pregnancy are increasingly treated endovascularly. Mode of aneurysm treatment does not affect mortality or discharge destination in pregnancy.

## Introduction

Cerebral aneurysms are focal arterial pathology found in 1.8% of women of reproductive age [1]. Rupture of these aneurysms results in an aneurysmal subarachnoid hemorrhage (aSAH), which occurs between 3 to 11 per 100,000 pregnancies and is associated with significant pregnancy-associated maternal and neonatal morbidity and mortality [2].

The physiological and anatomic maternal vascular adaptations of pregnancy can modulate the rate of aneurysmal growth and subsequent rupture. However, studies looking into increased rate of aSAH during pregnancy and delivery have had inconsistent results [3, 4].

The effect of pregnancy on the natural history of subarachnoid hemorrhage, functional outcomes, and the most appropriate treatment strategy for aSAH during pregnancy remain unclear. The possible impact of the type of treatment for securing the aneurysm, the course of critical care provided after hospitalization, and the obstetrical care, and timing of each of the treatment are unknown. We aim to study the effect of pregnancy on aSAH functional outcome and to investigate national trends of treatment utilization for cerebral aneurysms, and its association with functional outcome during pregnancy.

## Methods

We performed a retrospective observational cohort study using data from the largest United States all-payer inpatient claims-based database, the National Inpatient Sample (NIS) between the years of 2010 and 2018. Maintained by the Healthcare Cost and Utilization Project (HCUP), the NIS is the largest publicly available all-payer inpatient database in the United States (US) and samples 20% of all hospital discharges. Using robust survey-weighting algorithms, the NIS provides estimates for approximately 97% of all hospitalizations in the US. Due to the de-identified nature of the NIS, this study was deemed exempt from full review by the Institutional Review Board at our institution [5].

Hospitalization was identified according to the International Classification of Disease, the 9th and the 10th revision (ICD-9 and 10). All the hospitalization for women between the ages of 18 to 45 with diagnosis of subarachnoid hemorrhage (SAH) and concurrent aneurysmal treatment were extracted to ensure the etiology of the SAH was aneurysmal. Subsequently the subgroup of this cohort who had concurrent pregnancy related hospitalization were identified (S1 Table).

The discharge destination for the hospitalization was used as a surrogate for the functional outcome following aSAH. Functional outcome was dichotomized to "good" defined as discharge to home or home with services and "bad" defined as discharge to short or long-term care facility or death.

Multivariate analyses were performed to evaluate for the effect of pregnancy state on mortality and probability of good outcome while controlling for age, subarachnoid severity,

diagnosis of diabetes, hypertension, mode aneurysm treatment, hospital size, teaching status, and region.

Further multivariate analyses were performed to evaluate the effect of underlying pregnancy risk factors, such as hypertensive diseases of pregnancy, chronic hypertension and diabetes and effect of mode of treatment on outcome of aSAH in pregnancy related hospitalization while controlling for age, severity, hospital size, teaching status, and region.

For evaluation of severity of subarachnoid hemorrhage for each hospitalization, the previously validated NIS subarachnoid severity scale (NIS-SSS) was utilized. NIS-SSS is calculated as the weighted average of clinical characteristics which has been validated and outperformed other proposed scales in predicting the aSAH outcome using the NIS [6].

To investigate the trends of aneurysm treatment modality used in pregnancy the proportion of coiling and clipping treatments were broken down for each year in the general population and then cohort of hospitalizations associate with pregnancy. Further multivariate analyses were also performed to evaluate for the effect of baseline characteristics on the treatment modality used.

## Results

We identified 13,351 aSAH with treatment during that study interval, of which 440 were among pregnant people (Fig 1). The clinical characteristics of all and the pregnancy associated hospitalization are summarized in Table 1. Younger age was associated with the pregnancy cohort (32.3 vs 38.8 years old, p<0.001). The mortality rates for the pregnant and non-pregnant cohort were 6.8% and 8.2%, respectively. In the multivariate analysis controlling for all other factors, pregnancy status did not affect the mortality rate in aSAH (OR 0.8, 95% CI = 0.2–2.2, p = 0.57), nor did it affect the rate of "good" functional outcome (OR 1.5, 95% CI = 0.8–3.3, p = 0.20).

There was a significant association with higher mortality during pregnancy with higher subarachnoid severity (OR = 1.85, p = 0.04), chronic hypertension (OR = 12, p = 0.01) and smaller size of the hospital (OR = 3.75, p = 0.01). The effect of clinical characteristics on mortality with aSAH during pregnancy has been summarized in Fig 2A. There were also trends towards higher mortality with older age and hypertensive disease of pregnancy.

There was a significant association with subarachnoid severity and worse functional outcome during pregnancy. The results of the multivariate model evaluating the effect of clinical characteristics on rate of good functional outcome with aSAH during pregnancy has been summarized in Fig 2B.

When comparing treatment modalities in the pregnancy cohort, there was no significant difference in rate of endovascular treatment versus microsurgical treatment in compared to the non-pregnant cohort (p = 0.31). Like all comers in this age range the treatment of ruptured aneurysms in pregnancy are increasingly through endovascular approaches (Fig 3).

The multivariate model evaluating the clinical characteristics of pregnancy associated aSAH hospitalizations by treatment modality of the aneurysm is summarized in Fig 4. There was only significant association with diagnosis of hypertensive disease of pregnancy and endovascular treatment (OR = 3.5, p = 0.02).

## Discussion

Normal physiologic changes of pregnancy, such as altered cardiovascular hemodynamics, modulated coagulation profiles, and hormonal changes on vessel wall have been postulated to increase the risk of aneurysm growth and rupture [3, 7–10]. However, the validity of this theory remains to be proven. Such physiological changes would also affect the severity of the

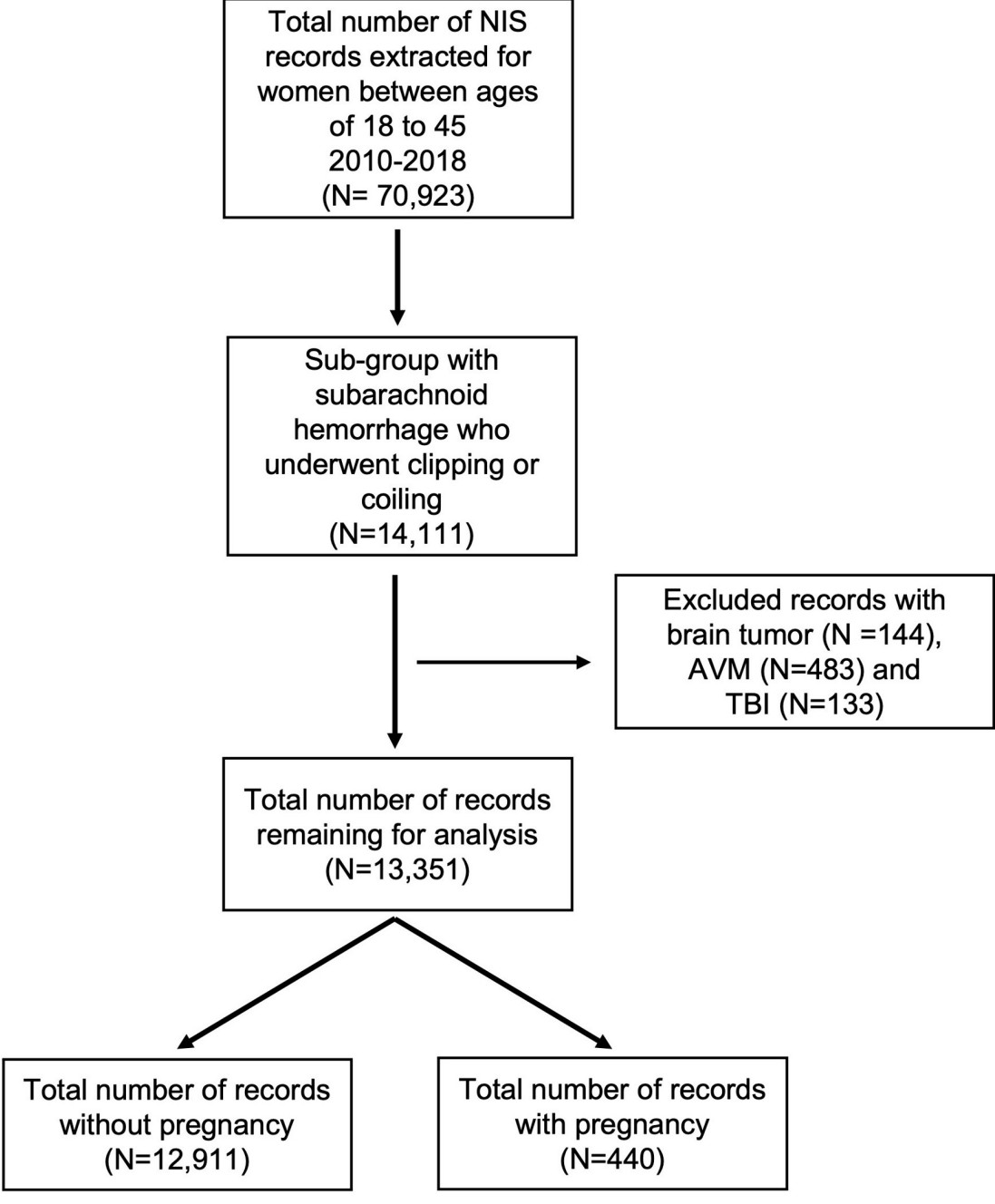

**Fig 1. Extraction of study population.** This figure illustrates the manner in which records were extracted from the NIS data set. It resulted in total of 13,351 records with aneurysmal subarachnoid hemorrhage 440 of which were associated with pregnancy.

brain damage at the ictus of hemorrhage as well as altering the course of the cascade pathologies to follow. Using the data set representing the current state of practice in the United States over the last decade, in this study we demonstrate that pregnancy does not affect the rate of functional outcome or mortality of aneurysmal subarachnoid hemorrhage while controlling for other clinical factors. Clinical severity of the presentation after the ictus also is comparable

**Table 1. Baseline characteristics of patients hospitalized with aneurysmal subarachnoid hemorrhage with pregnancy and without pregnancy.**

| Variable | Non-pregnant persons (n = 12,911) | Pregnant persons (n = 440) | p-value |
|---|---|---|---|
| Age; median (Q1-Q3) | 38.8 (33.0–42.3) | 32.3 (27.7–38.0) | < .01 |
| NIS-SSS; median (Q1-Q3) | 0.9 (0–2.0) | 0.8 (0–1.8) | 0.4640 |
| Hypertensive Disease of Pregnancy | | 71 (16.2%) | |
| Chronic Hypertension | 6886 (53.3%) | 184 (41.8%) | 0.04 |
| Diabetes Mellitus | 780 (6.0) | 20 (4.4%) | 0.53 |
| Smoking | 4824 (37.4%) | 132(30.0%) | 0.17 |
| **Treatment** | | | 0.42 |
| Coil | 9048 (70.1%) | 326 (74.0%) | |
| Clip | 3863 (29%) | 115 (26.0%) | |
| **Hospital bedsize** | | | 0.75 |
| Missing | 120 (0.9%) | . | |
| Small | 528 (4.1%) | 13 (2.9%) | |
| Medium | 1795 (13.9%) | 74 (16.9%) | |
| Large | 10468 (81.1%) | 353 (80.2%) | |
| **Hospital Region** | | | 0.47 |
| Northeast | 2267 (17.6%) | 70 (15.8%) | |
| Midwest | 2806 (21.7%) | 69 (15.8%) | |
| South | 5141 (39.8%) | 195 (44.2%) | |
| West | 2697 (20.9%) | 106 (24.2%) | |
| **Hospital Teaching status** | | | 0.39 |
| Missing | 120 (0.9%) | . | |
| Rural | 39 (0.3%) | . | |
| Urban nonteaching | 1046 (8.1%) | 44 (10.0%) | |
| Urban teaching | 11706 (90.7%) | 396 (90.0%) | |

in the cohorts with and without pregnancy using NIS_SSS, a previously validated measure of clinical severity.

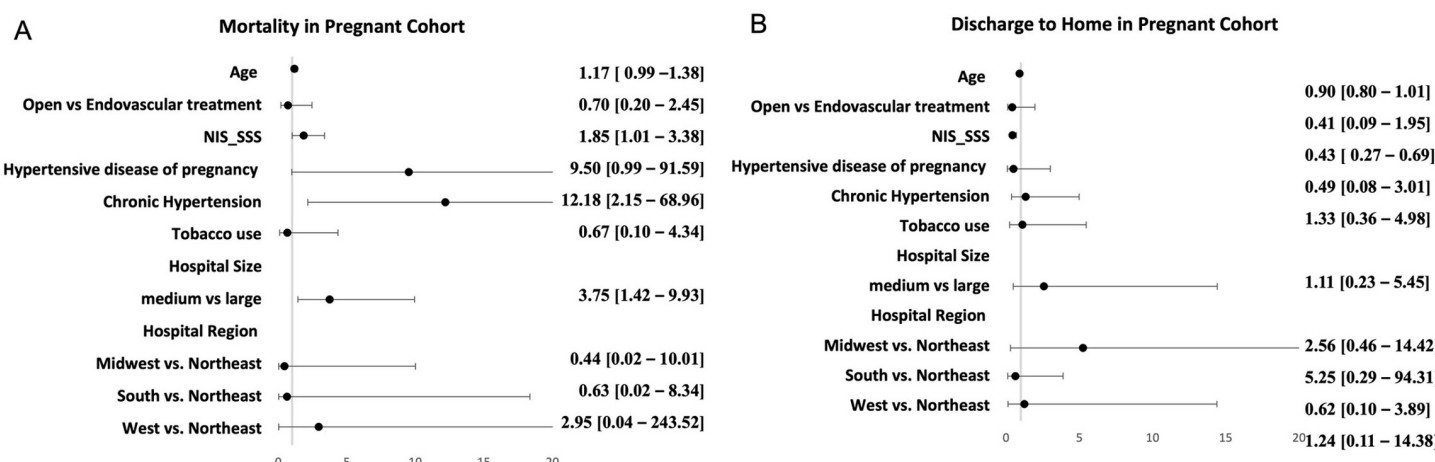

**Fig 2.** Multivariate model of patient characteristics associated with A. mortality and B. discharge to home in aneurysmal subarachnoid hemorrhage with pregnancy.

## Trend of intervention utilization in ruptured aneurysms

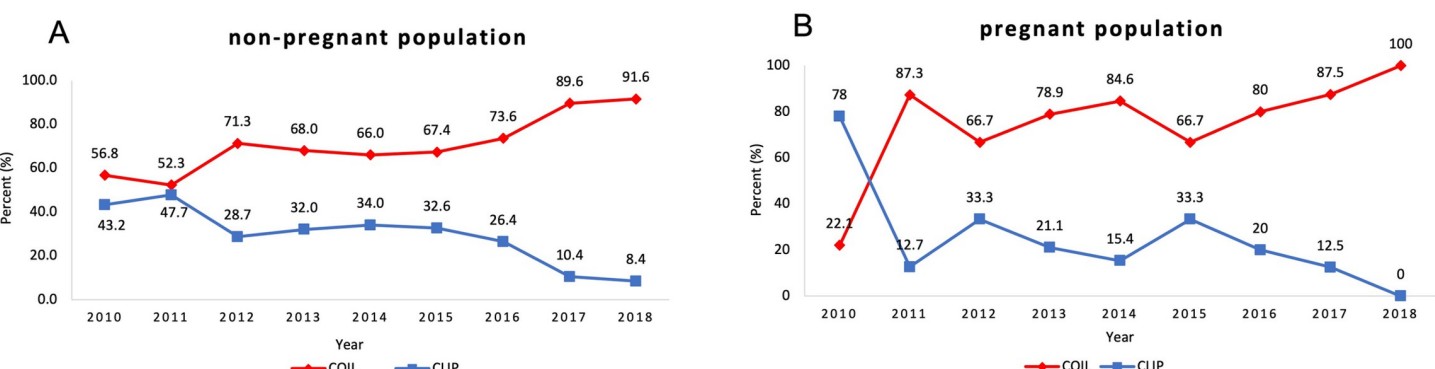

**Fig 3.** Shows trends of mode of aneurysm treatment after subarachnoid hemorrhage in the hospitalizations associated A. without and B. with pregnancy.

Like the general population NIS_SSS is associated with significantly higher rate of mortality and lower rate of good functional outcome in the pregnant cohort. There is also strong trend for association of age with higher mortality and worse functional outcome in pregnancy despite all the patients being younger age range relatively. The other variable significantly associated with higher rate of mortality in the pregnancy is diagnosis of chronic hypertension, which is unlike the observation in the general population [11]. Hypertension has been shown to be a possible protective factor after aSAH in the general population. It is thought to be secondary to lack of reactivity of the arterial tree due to chronic remodeling which would down modulate cerebral vasospasm and delayed cerebral ischemia. Given the young age of the cohort of patients being studied there would be minimal remodeling of the vessel wall from

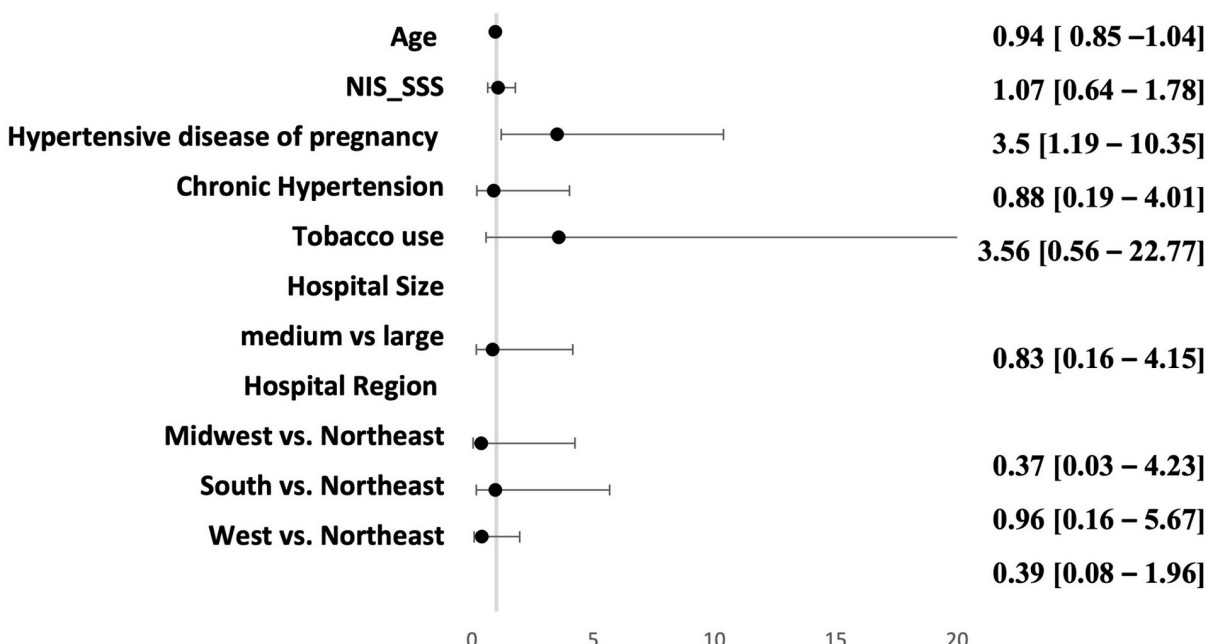

**Fig 4. Multivariate model of patient characteristics associated endovascular treatment vs clipping in aneurysmal subarachnoid hemorrhage associate with pregnancy.**

hypertension. Additionally, there is a strong trend with diagnosis of hypertensive disease of pregnancy and higher rate of mortality from aneurysmal subarachnoid hemorrhage. This set of disorders are independently a leading cause of maternal mortality worldwide [12].

Specifically, our data demonstrates that low-volume hospitals were associated with higher maternal mortality with aSAH. Pregnant people with high-risk conditions, such as aSAH, benefit from birthing in centers with multi-disciplinary subspeciality services. Caring for these high-risk individuals at high acuity centers have been associated with improved outcomes [13]. Pregnant people with high comorbidities have a higher adjusted relative risk of severe maternal morbidity birthing in low acuity versus high-acuity hospitals compared to (adjusted OR, 9.55; 95% CI, 6.83–13.35 vs 6.50; 95% CI, 5.94–7.09) [14].

The best approach to treatment of aneurysm remains unclear in pregnancy. However, given the life-threatening nature of the aneurysmal subarachnoid hemorrhage, treatment of ruptured aneurysms should be performed promptly and take priority over obstetrical concerns [10, 15, 16]. Optimizing maternal health, and hence early treatment of ruptured aneurysm improves both maternal and fetal outcome [16, 17]. However, if the pregnant person is term or late preterm, there is a debate if delivery should take precedence over treatment of the aneurysm, and a multi-disciplinary discussion should take place individualizing that person's care. Some studies suggest delivery before treatment would theoretically decrease anesthetic and potential procedure related morbidity to the fetus, but focus on the fetus in the dyad [18, 19], and other studies suggest delivery immediately following aneurysm treatment with concern for high risk of early aneurysmal rupture specially in the setting of marked hemodynamic changes immediately postpartum with large volume of autotransfusion [20, 21].

Clipping and endovascular treatment have been shown to be safe and effective in pregnancy in limited case series [22, 23]. However, both treatments methods encompass procedural morbidities which could potentially be more harmful during pregnancy. Some studies have expressed concerns about extrapolating from International Subarachnoid Aneurysm Trial to subgroup of patients with pregnancy [24, 25]. The most immediate concern with endovascular treatment during pregnancy is the radiation exposure to the fetus. However, this exposure is not significant and can be further reduced with shielding the abdomen and pelvis with double lead apron, use of advanced fluoroscopy, decreasing the frame rate and collimation of radiation [26]. Use of radial approach eliminates the need for direct pelvic radiation used for femoral access and approach [27]. In a phantom study to assess the risk of coil embolization during pregnancy measuring typical absorbed fetal dose to range from 0.17 to 2.8mGy. Such low level of irradiation exposes the fetus to orders of magnitude less than natural frequency of heritable diseases or natural cumulative risk of fatal childhood cancers [28]. The other concern is for risk for regrowth and re-rupture during pregnancy in the setting of residual aneurysm after endovascular treatment [29]. Need for antithrombotic agents during and or after endovascular treatment is another concern [30]. Clipping an aneurysm during pregnancy is also not risk free, and requires deep sedation, low blood pressure and hyperventilation which could potentially be dangerous to the fetus [18]. Additionally, there has been higher rates of symptomatic vasospasm, cardiomyopathy and re-bleeding reported in this cohort [8].

In this study the mode of treatment of ruptured aneurysm doesn't significantly affect mortality or functional outcome during pregnancy. During the study interval 75% of ruptured aneurysms during pregnancy were treated endovascularly. Which is very different compared to the practice from the prior to decades where 77% of ruptured aneurysms during pregnancy were treated by clipping [17]. Looking at the trend of treatment over the studied decade shows evolution of practice with progressively more ruptured aneurysms during pregnancy being treated endovascularly. This is similar to the trend of treatment in the non-pregnant cohort within the same age range.

This study has several limitations. The primary limitation of this study is the relatively small sample size. Despite using the NIS which is an enormous data base low incidence of aneurysmal rupture in pregnancy limits the number patients studied within this time interval. This results in reduced statistical power in analyzing the effect of potentially relevant clinical variables. However by using the population and its break down within the united states 2021 and the observed prevalence of aSAH in women between the ages of 18 and 45 estimated number of patients would be 32480 individuals. Therefore this study including 13351 patients represents less 50% of this population and would potentially limit the generalizability of the findings.

To make sure that all subarachnoid patients captured from NIS are secondary to aneurysmal rupture we included having undergone clipping or endovascular treatment as an inclusion criterion. This criterion will omit the patients who did not receive treatments or did not survive to undergo treatment. This omission could potentially affect the conclusion about the effect of pregnancy on outcome of aSAH. The recent systemic review of patient with pregnancy and subarachnoid hemorrhage with respectively 9 and 22% of the included patients not receiving any treatments report mortality rates of 11 and 20% which is markedly higher than our observation.

Based on the current available clinical variables in NIS the best measure of clinical outcome is the discharge destination which is non specific and does not directly present the extent of neurological function or lack there off. Given the data available in NIS is limited to hospitalizations we are unable to study long term clinical outcome and can only study the outcome of the hospitalization. Additionally given the limitation of the database we are unable to identify the gestation age and stage of pregnancy at which point the aSAH occurred and unable to evaluate the out of the labor and the functional out of the offspring.

## Conclusion

Pregnancy does not alter the clinical outcome or mortality from aSAH. The mode of aneurysmal treatment after aSAH during pregnancy does not affect mortality or rate discharge to home. Cerebral aneurysms after rupture during pregnancy are progressively being treated through endovascular approach in the last decade. Mode of aneurysm treatment and the timing should be guided by a multidisciplinary team including vascular neurosurgery, neurointerventional, and maternal fetal medicine in high-acuity hospital settings.

## Supporting information

**S1 Table. ICD9 and ICD 10 diagnostic and procedure codes used to extract the study population and the subgroups from NIS.**
(DOCX)

## Author Contributions

**Conceptualization:** Kasra Khatibi, Yalda Afshar.

**Data curation:** Kasra Khatibi, Hamidreza Saber.

**Formal analysis:** Kasra Khatibi, Hamidreza Saber, Smit Patel.

**Methodology:** Kasra Khatibi, Hamidreza Saber, Lucido Luciano Ponce Mejia, Naoki Kaneko, Viktor Szeder, May Nour, Reza Jahan, Satoshi Tateshima, Geoffrey Colby, Gary Duckwiler, Yalda Afshar.

**Writing – original draft:** Kasra Khatibi, Yalda Afshar.

**Writing – review & editing:** Hamidreza Saber, Lucido Luciano Ponce Mejia, Naoki Kaneko, Viktor Szeder, May Nour, Reza Jahan, Satoshi Tateshima, Geoffrey Colby, Gary Duckwiler, Yalda Afshar.

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
