## [Decision Letter · Decision Letter 0]

23 Jan 2023

PONE-D-22-33076Aneurysmal Subarachnoid Hemorrhage in Pregnancy: National Trends of Treatment, Predictors, and OutcomesPLOS ONE

Dear Dr. Khatibi,

Thank you for submitting your manuscript to PLOS ONE. After careful consideration, we feel that it has merit but does not fully meet PLOS ONE’s publication criteria as it currently stands. Therefore, we invite you to submit a revised version of the manuscript that addresses the points raised during the review process.

We look forward to receiving your revised manuscript.

Kind regards,

Devi Prasad Patra, MD, MCh, MRCSEd

Academic Editor

PLOS ONE

Journal Requirements:

Additional Editor Comments:

This is a well written manuscript discussing an important topic of effect of pregnancy on outcome in patients with subarachnoid hemorrhage. The reviewers have provided important questions and suggestions that i would like the authors to address.

Reviewers' comments:

Reviewer's Responses to Questions

**Comments to the Author**

1. Is the manuscript technically sound, and do the data support the conclusions?

Reviewer #1: Yes

Reviewer #2: Partly

Reviewer #3: Yes

Reviewer #4: Yes

2. Has the statistical analysis been performed appropriately and rigorously? 

Reviewer #1: Yes

Reviewer #2: No

Reviewer #3: I Don't Know

Reviewer #4: Yes

3. Have the authors made all data underlying the findings in their manuscript fully available?

Reviewer #1: Yes

Reviewer #2: Yes

Reviewer #3: Yes

Reviewer #4: Yes

4. Is the manuscript presented in an intelligible fashion and written in standard English?

Reviewer #1: Yes

Reviewer #2: No

Reviewer #3: Yes

Reviewer #4: Yes

5. Review Comments to the Author

Reviewer #1: This is a well written article on the topic - aneurysmal SAH in pregnancy.

The authors included the National Inpatient Sample from 2010-2018, and identified all birth hospitalizations of women between ages of 18 to 45 associated with subarachnoid hemorrhage and aneurysm treatment.

2 comparative groups were generated - 440 aSAH patients with pregnancy, and 12911 aSAH nonpregnant patients.

They showed an important message - "Pregnancy does not alter mortality or the discharge destination for aSAH."

One of the limitation as stated by the authors was "the relatively small sample size". It would be important for the authors to estimate the total numbers of aSAH between 18-45 age group over the same time period (2010-2018), and what the 13351 patients used in this study contribute to the overall disease burden.

Using the incidence of aSAH as 10/100000, in the USA with 330million population that would be 33 000 aSAH per year. Over the 8 year period (2010-2018), there would be 264 000 aSAH. If we assumed that 20-30% of aSAH are in the 18-45 year old, that would be an estimate of 52800 - 79200.

So the study with 13351 aSAH patients might represent 16-25% of the whole cohort from the USA over similar timeline, which is pretty good for a study. At least if it were clear in the limitation that the conclusion from the study was probably based on less than 25% of the true cohort, it is still a well balanced study.

Reviewer #2: It remains unclear whether this study has sufficient statistical power; please analyse and discuss.

Treatment of ruptured aneurysms after poor grade aneurysmal subarachnoid hemorrhage (aSAH), i.e., hemorrhages with Hunt and Hess (H&H) grade 5 (and in some cases also grade 4), remains controversial. Since untreated ruptured aneurysms have not been included in this study, there is probably a high selection bias in favor of aSAH with H&H grades 1-3; please discuss adequately. There are several recent studies on outcome after interdisciplinary treatment for aSAH which should be included in the discussion.

Reviewer #3: the authors present a database retrospective analysis of pregnant patients compared to subarachnoid hemorrhage non pregnant females they find no significant difference in outcomes with the admittedly broad non specific tool of discharge destination as a surrogate for outcome.

Reviewer #4: This is a retrospective analysis of the National Inpatient Sample looking at subarachnoid hemorrhage in pregnant women. The manuscript is well written and the analysis is appropriate. Their main insight is that more pregnant patients are being treated via endovascular approaches than in past. This is in line with modern practice in general.

The challenges in NIS data analyses are well known and well documented. At best, these studies suggest correlations that may be drawn but there is no causation and the limited amount of granular detail means that the clinically applicable insights are few and far between. The authors do a good job of outlining the implications of their observations but they could emphasize in the Discussion some of these limitations.

Otherwise, the manuscript is well written and would be appropriate for publication.

6. PLOS authors have the option to publish the peer review history of their article (what does this mean?). If published, this will include your full peer review and any attached files.

Reviewer #1: No

Reviewer #2: No

Reviewer #3: **Yes: **Matthew J Koch

Reviewer #4: No

---

## [Author Response · Author response to Decision Letter 0]

28 Mar 2023

Response to Reviewer #1: 

Thank you for pointing out this limitation and the granular analysis. Using the break down of the population by age published from the year 2021 with estimating the total population to be 331.89 million, the proportion of the women between the ages of 18 through 45 was estimated to be 58million people which is likely a slight over estimation compared to this population between the years of 2012 to 2018 given the increase in population(1). De Rooj et al in their international systematic review estimated the incidence of aneurysmal subarachnoid hemorrhage for women between the ages of 18-45 to be around 7 per 100,000 person year.(2) Using this estimation the incidence of aSAH within the women in this age group over this time period would be estimated to be 32480. So the study cohort with 13351 patients is representative of less than 50% of the true cohort. This limitation will affect the generalizability of the conclusion to the general united states population. According changes were made to the manuscript to elaborate on this limitation. 

Response to Reviewer #2: 

Thank you for pointing out this limitation in inclusion criteria which has definitely confounded the selection and therefore the observed outcome. To ensure that the underlying etiology of subarachnoid hemorrhage in the included patients were aneurysmal we limited the inclusion criteria to the patients who have had open or endovascular surgical treatment. Based on the coding used including all subarachnoid patients specially in young women and young pregnant women would potentially include a large proportion of non-aneurysmal subarachnoid hemorrhages. 

However, as it was suggested by not including patients with aneurysmal subarachnoid hemorrhage who did not undergo treatment could potentially exclude the patients who had devastating neurological disability after the ictus of hemorrhage. By excluding all or proportion of such patients could potentially observe high rate of good function of outcome and lower mortality rate as mentioned in the limitation paragraph of the discussion section. 

As again suggested by the reviewer in the recently published pooled case series and systematic reviews show a higher rate of mortality compared to our observation (6.8%). Robba et al. in their pooled review of 52 patients 9% of whom did not receive any treatments with reported mortality rate of 11%. (3) Beighley et al in a more recent systemic review of 54 patients 22% whom did not undergo treatment report mortality rate of 20%. (4) According changes were made to the discussion section to further elaborate on this limitation. 

Thank you for the comment about the power of the study. We agree that with a prospective study, power analysis would significantly help study design and conclusions. However, we are utilizing a retrospective cohort and because of the study design which demonstrates likely no difference in between the effect size for mortality and functional outcome between the studied cohort post hoc power calculation would not likely be helpful be helpful.(5-7) Specially given the post hoc power analysis is to identify that the “non-significant” hypothesis test failed to achieve significance as it was not powerful enough and requiring higher degrees of freedom and a larger sample size. Though the number of subjects remain limited in this study it is performed using one of the larger openly available database of patients in the united states. The largest pooled series and systemic review of aneurysmal subarachnoid hemorrhage in pregnancy have included upto 50 patients which shows the infrequency of such cases and difficulty with obtaining a larger sample size. 

Response to Reviewer #3:

Thank you for pointing out the non specificity of the criteria for measurement of functional outcome. Given the limitations of use of NIS database discharge destination was the more reliable variable that we could utilize for comparison of these cohorts. According changes were made to the discussion section to emphasize this limitation. 

Response to Reviewer #4:

Thank you for pointing out the limitation of NIS based analyses. As described in the discussion section and further pointed out by the other reviewers use of NIS has led to need for use of more strict inclusion criteria which affects the generalizability of the observations. Additionally given the limitations the measure used for evaluation of functional outcome is non specific and based only the condition at discharge with no long term outcome available. The manuscript to further emphasize such short comings.

1. Duffin E. Resident population of the United States by sex and age as of July 1, 2021 statista.com2022 [

2. de Rooij NK, Linn FH, van der Plas JA, Algra A, Rinkel GJ. Incidence of subarachnoid haemorrhage: a systematic review with emphasis on region, age, gender and time trends. J Neurol Neurosurg Psychiatry. 2007;78(12):1365-72.

3. Robba C, Bacigaluppi S, Bragazzi NL, Bilotta F, Sekhon MS, Bertuetti R, et al. Aneurysmal Subarachnoid Hemorrhage in Pregnancy-Case Series, Review, and Pooled Data Analysis. World Neurosurg. 2016;88:383-98.

4. Beighley A, Glynn R, Scullen T, Mathkour M, Werner C, Berry JF, et al. Aneurysmal subarachnoid hemorrhage during pregnancy: a comprehensive and systematic review of the literature. Neurosurg Rev. 2021;44(5):2511-22.

5. Althouse AD. Post Hoc Power: Not Empowering, Just Misleading. J Surg Res. 2021;259:A3-A6.

6. Goodman SN, Berlin JA. The use of predicted confidence intervals when planning experiments and the misuse of power when interpreting results. Ann Intern Med. 1994;121(3):200-6.

7. Hoenig JM HD. The abuse of power: the pervasive fallacy of power calculations for data analysis. . Am Stat 2001;55:19-24.

---

## [Decision Letter · Decision Letter 1]

16 Apr 2023

Aneurysmal Subarachnoid Hemorrhage in Pregnancy: National Trends of Treatment, Predictors, and Outcomes

PONE-D-22-33076R1

Dear Dr. Afshar,

We’re pleased to inform you that your manuscript has been judged scientifically suitable for publication and will be formally accepted for publication once it meets all outstanding technical requirements.

Kind regards,

Devi Prasad Patra, MD, MCh, MRCSEd

Academic Editor

PLOS ONE

Additional Editor Comments (optional):

Thank you for addressing all the queries. Reviewers are now satisfied with the revised version for publication.

Reviewers' comments:

Reviewer's Responses to Questions

**Comments to the Author**

Reviewer #2: All comments have been addressed

Reviewer #3: All comments have been addressed

2. Is the manuscript technically sound, and do the data support the conclusions?

Reviewer #2: Yes

Reviewer #3: Yes

3. Has the statistical analysis been performed appropriately and rigorously? 

Reviewer #2: Yes

Reviewer #3: Yes

4. Have the authors made all data underlying the findings in their manuscript fully available?

Reviewer #2: Yes

Reviewer #3: Yes

5. Is the manuscript presented in an intelligible fashion and written in standard English?

Reviewer #2: Yes

Reviewer #3: Yes

6. Review Comments to the Author

Reviewer #2: Authors have submitted a revised version of their manuscript. The concerns of the reviewer have been addressed adequately.

Reviewer #3: The authors have addressed all raised comments. No further commentary at this point regarding the manuscript.

7. PLOS authors have the option to publish the peer review history of their article (what does this mean?). If published, this will include your full peer review and any attached files.

Reviewer #2: No

Reviewer #3: No

---

## [Editor Report · Acceptance letter]

25 Apr 2023

PONE-D-22-33076R1 

Aneurysmal Subarachnoid Hemorrhage in Pregnancy: National Trends of Treatment, Predictors, and Outcomes 

Dear Dr. Afshar:

I'm pleased to inform you that your manuscript has been deemed suitable for publication in PLOS ONE. Congratulations! Your manuscript is now with our production department. 

Kind regards, 

on behalf of

Dr. Devi Prasad Patra 

Academic Editor

PLOS ONE